# Spatial Augmented Reality User Interface for Assistive Robot Manipulation

Alexander Wilkinson, Amelia Sinclaire, and Holly A. Yanco
{alexander_wilkinson,amelia_sinclaire}@student.uml.edu,holly@cs.uml.edu
University of Massachusetts Lowell
Lowell, MA, USA

## ABSTRACT

We present the design of a spatial augmented reality (SAR) system for an assistive robot manipulator mounted to a mobility scooter. Our system is capable of directly designating objects for the system to grasp using a cursor projected into the manipulator's workspace that can be manipulated with a joystick. It also displays the manipulator's workspace by highlighting surfaces within the reach of the manipulator, and communicating robot intent through highlighting segmented objects that the system intends to grasp. In this paper, we describe current work on the design and implementation of this system, and a user study that is currently in progress that is intended to compare a system implemented using this SAR modality against a more traditional graphical user interface (GUI).

**ACM Reference Format:**
Alexander Wilkinson, Amelia Sinclaire, and Holly A. Yanco. 2023. Spatial Augmented Reality User Interface for Assistive Robot Manipulation. In *Proceedings of VAM-HRI '23.* ACM, New York, NY, USA, 5 pages.

## 1 INTRODUCTION

In our prior work [12, 26, 28], we have implemented several designs for a *direct selection* system used with an assistive robot manipulator. These systems enable a user to designate an object for the robot to pick by directly manipulating a laser or beam of light to point to the desired object.

Kemp et al. [20] proposed an interface that utilized a laser pointer for a user to designate a location in 3D space. The interface detects a laser using a system of cameras to estimate the 3D position of the laser dot. It was found that this interface allowed users to easily and robustly communicate 3D locations. In our prior work [12, 26], we implemented a similar laser selection method for a grasping system on a mobility scooter equipped with a robot arm. Our system is intended to assist people with limited mobility in grasping objects. Our experiments demonstrated that laser selection was a viable method for object designation in our system.

Direct selection with a laser has several limitations. First, if the position and orientation of the laser pointer are not precisely known then the system must rely on detecting the laser with cameras. Although the use of servo motors or other precise types of rotary actuators can allow the system to estimate the pose of the laser

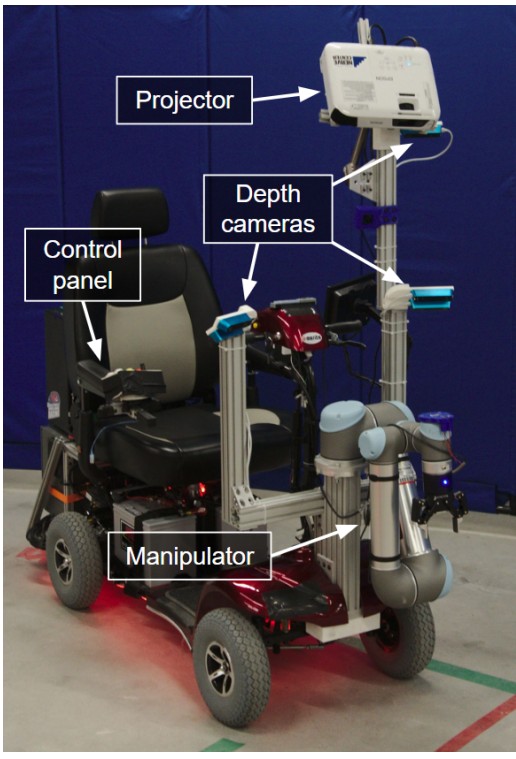

**Figure 1: Our assistive robot manipulator mounted to a mobility scooter.**

diode, a very small amount of rotational error in the measurement of the rotation of the diode can compound over distance into a large amount of translational error in estimating the position of the laser dot. It is therefore necessary to estimate the position of the laser using calibrated cameras. This introduces the possibility of false positives or negatives in laser dot detection, especially when there is varying lighting and surface reflectivity.

Second, the laser as a UI modality has inherent limitations in communicating robot intent as it is only capable of projecting a single dot of constant color and brightness. With only a single laser dot, it is not possible to highlight an area or draw symbols or text in the environment.

We have developed a user interface (UI) that builds upon the capabilities of laser-based direct selection methods by utilizing a spatial augmented reality (SAR) system consisting of a projector and multiple depth cameras. Our system is capable of performing direct selection using a "cursor" projected into the real world

that functions similarly to a laser dot. Additionally, it is capable of displaying the robot's workspace directly in the real world, and indicating robot intent by highlighting objects it intends to grasp. The system can also communicate using a variety of colors at varying brightness levels.

The SAR system can be thought of as "inside-out," where the visual information being displayed directly represents the belief of the system. This may be more robust than an "outside-in" system which must detect the position of a laser through external sensing. Additionally, the increased capability of the SAR modality may enhance the communication ability of our system.

## 2 RELATED WORK

SAR modalities are increasingly being investigated in the field of human-robot interaction (HRI) as a means to communicate spatial information such as robot intent and improve human-robot collaboration.

Researchers have investigated using SAR modalities to communicate the intent of robots. Some of these focus on communicating the intent of robotic manipulators. Ganesan et al. presented a paradigm for promoting human-robot collaboration with a robotic manipulator for assembly tasks by using the work environment as a canvas to project visual cues communicating instructions and robot intent [10]. Sonawani and Amor proposed a framework to communicate the intent of a manipulator by highlighting objects with a projector and displaying a shadow of the manipulator that executes the manipulator's trajectory before the manipulator moves [25]. A number of approaches have also been proposed to allow mobile robots to communicate navigation intent with SAR using simplified maps and arrows [6, 7, 19, 27]. In our prior work, we extended the SAR capabilities of a mobile manipulation system to enable projecting navigation paths onto the floor and created an open-source implementation to allow this system to be implemented on other robots [14].

Gelšvartas et al. proposed a projection mapping UI for assistive object selection utilizing a depth camera and a projector that is capable of highlighting objects in the environment [11]. This interface allows the user to select from a list of detected objects by highlighting each object and allowing the user to either select the current object or move to the next object.

Our work extends prior and related work in SAR modalities to a UI for an assistive robotic manipulator, motivated by the need for an accessible and easy-to-learn UI for novel users who might have limited mobility, fine motor control, or vision.

## 3 SYSTEM OVERVIEW

### 3.1 Mobility scooter

Our mobility scooter platform, developed in prior work [26], and shown in Figure 1 consists of a Universal Robots UR5 robot arm mounted to the front of a Merits Pioneer 10 mobility scooter. It is intended to assist people with limited mobility in picking and placing objects in an open environment such as a grocery store. Five Structure depth cameras are used for perception, and the software is implemented in ROS [22].

### 3.2 Hardware

Our SAR UI consists of a projector, a control panel, and a pair of speakers. The projector is an Epson Home Cinema 640 which has a brightness of 3200 lumens and a resolution of 800 x 600. It is mounted on the scooter above the manipulator's workspace, pointing down.

The control panel, shown in Figure 3, is a 3D printed enclosure with a rotary switch, a joystick, and five buttons with internal LEDs, all connected to an Arduino Uno. The Arduino Uno uses `rosserial` to publish the state of the switch, the joystick, and the buttons. It also subscribes to a message that contains the desired state of each of the lights, which can light up to indicate what actions can be taken by the user in a given state.

Two speakers mounted near the headrest of the scooter allow the system to utilize voice prompts to guide the user. The volume can be adjusted by the user.

### 3.3 Projection mapping

SAR is provided by a projection mapping system that can project information onto surfaces in the environment. Our projection mapping system uses the principle that a projector is the dual of a camera to project light onto specific 3-dimensional coordinates in the real world. Our approach to implementing projection mapping is described in greater detail in our prior work [15, 26, 28] and is similar to [11].

Visualizations for the virtual laser, workspace display, and intent display are rendered in RViz, a 3D visualization tool for ROS. The image sent to the projector is generated by the `rviz_camera_stream` RViz plugin, which publishes a rendered camera view as a `sensor_msgs/Image` message based on intrinsics defined by a `sensor_msgs/CameraInfo` message and extrinsics defined by a TF [9] transform.

An overview of the individual components of our projection mapping system follows:

(1) Five depth cameras capture the 3D geometry of the manipulator's workspace. The pointclouds captured by these five depth cameras is merged into a single pointcloud.
(2) Using this geometry as a basis, operations are performed on the pointcloud to select areas of interest to be displayed. For object designation, points near a cursor controlled by the user are selected. For object intent communication, points that the system believes belong to the designated object are selected. For displaying the manipulator's workspace, points that are believed to be within reach of the manipulator are selected.
(3) A 3D visualization of the selected points is created in RViz. A virtual camera is defined in this visualization environment to render a 2D image to be displayed by the projector. The virtual camera is placed in the same pose as the projector, and uses the same intrinsics as the projector.
(4) The projector displays the rendered image. Using the principle that a projector is the dual of a camera, any visualizations made on surfaces in RViz are projected back onto the corresponding surfaces in the physical world.

The intrinsics of the projector lens are calculated manually by temporarily placing a flat calibration surface orthogonal to the

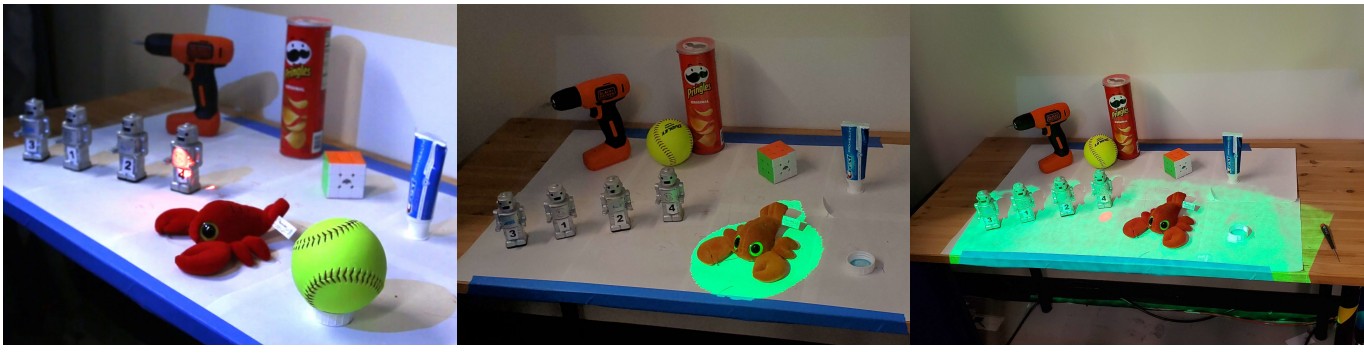

(a) object designation       (b) intent communication       (c) workspace display

**Figure 2: The system showing (a) object designation, (b) intent communication, and (c) workspace display. In (a), the user has placed the cursor, which is represented as red circle, over an object to designate it for grasping. In (b), an object is highlighted in green to communicate which object the robot intends to grasp. In (c), the workspace of the manipulator is displayed, where surfaces highlighted in green are within the reach of the manipulator.**

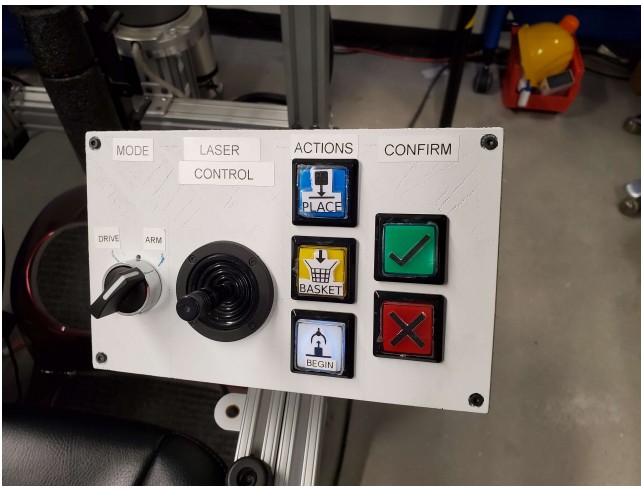

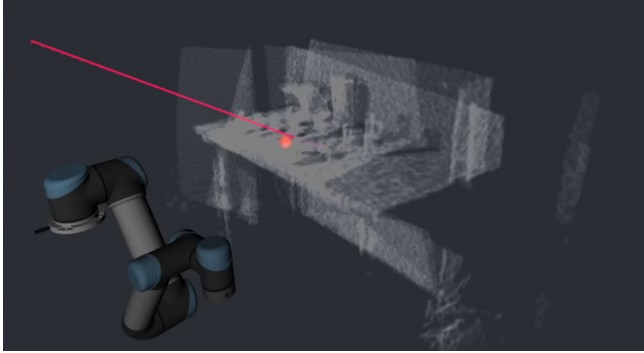

**Figure 4: A visualization of the object designation algorithm. A ray is cast with the origin centered approximately near the user's head, which they can rotate using the joystick. The cursor is placed at the intersection of the ray and the world as seen by the depth cameras.**

**Figure 3: The control panel for the SAR UI. The cursor is controlled by a joystick, and the buttons can light up to indicate what actions can be taken from a given state. Audio prompts are played through speakers that guide the user through using the interface.**

projector's optical axis, and taking measurements of the geometry of the projector's image – namely, the locations of the corners of the image. Using a pinhole model, we can calculate the focal length of the lens and the principal point of the lens, which can be placed into a camera matrix and projection matrix for the virtual camera. Distortion is assumed to be negligible, since consumer projectors are typically designed to produce a rectilinear image.

*3.3.1 Object designation.* Figure 2(a) shows the cursor placed over an object to designate it. The user's experience of the direct object designation is similar to that of a laser selection system. From the approximate location of the user's head when seated in the scooter, a ray is cast through the 3D environment. The direction the ray is

pointing is manipulated by the user with the joystick. A subset of points within a certain distance from the ray is selected, and the centroid of these points is calculated. The cursor is projected at the location of the centroid, similarly to how a laser would project a dot where the beam intersects with an object. Figure 4 shows a depiction of this ray intersecting the pointcloud with a cursor placed at the centroid.

The cursor is displayed in RViz as a `sensor_msgs/PointCloud2` message with a single point and subsequently projected onto the physical workspace.

*3.3.2 Intent communication.* Our system can communicate which object it intends to grasp, and its belief about how that object is segmented from the surrounding objects by highlighting an object. Figure 2(b) shows an object being highlighted in green light to communicate that it intends to grasp that object. Our segmentation algorithm returns a pointcloud that consists of only points that

belong to what it believes is the intended object, and these points can then be passed to the SAR system to be projected on top of the real-world object.

The segmented pointcloud that represents an object is displayed in RViz as a `sensor_msgs/PointCloud2` message, where the highlighted points are considered to belong to what the system believes is the designated object. These points are subsequently projected onto the physical workspace.

*3.3.3 Workspace display.* Figure 2(c) shows the workspace of the manipulator being displayed by the system, where the surfaces highlighted in green are within its workspace. The robot's workspace is defined as a radius from the robot's base as defined by the UR5 technical specifications. All points within this radius are then displayed in the SAR system.

The pointcloud that represents points in the robot's workspace is displayed in RViz as a `sensor_msgs/PointCloud2` message, where all of the displayed points are within the robot's workspace. These points are subsequently projected onto the physical workspace.

## 4 CURRENT WORK: HUMAN SUBJECT STUDY

Our SAR UI allows the user to keep their focus in the physical world when identifying and designating objects, which may have certain accessibility benefits. We are in the process of conducting a user study to validate our SAR UI and compare it to a more traditional graphical user interface (GUI) based entirely on a touchscreen. The GUI shows the user a picture of the robot's workspace on a touchscreen, and allows the user to tap on an object in order to designate it for selection.

We hypothesize that there is a relationship between a participant's spatial reasoning ability and their performance with or preference for each type of UI. This hypothesis follows from our observation that when using the GUI, the user must switch their focus between the picture of the world and the world itself. For example, if the user visually identifies an object in the world, the user must then locate the same object on the screen. This may cause additional cognitive load, especially in scenes with high amounts of visual clutter.

To test this hypothesis, we are conducting a within-subjects user study, recruiting participants 60 years of age or older since this demographic is more likely to be assisted by a mobility scooter. In our experiment, each participant is exposed to 4 conditions, combining the two UIs (GUI and SAR UI) and two levels of visual clutter (low-clutter and high-clutter) in the environment. Within each condition, the user is tasked with using the interface to select 15 objects identified by the experimenter with no previous knowledge of how to operate the interface. The experimenter points at each object using a pointer stick without naming the object to ensure that the participant must necessarily first visually identify the object in the real world. The order of conditions is randomized to control for learning effects and fatigue. We are collecting performance data such as task completion time and task success rate. We are evaluating the usability of each condition using the System Usability Scale (SUS), a tool designed to measure usability that consists of 10 Likert scale questions [5].

The spatial reasoning ability of participants is being measured using a modified version of the Revised Purdue Spatial Visualization Test (Revised PSVT:R): Visualization of Rotations, a psychometric instrument designed to measure spatial visualization ability using mental rotation of 3-dimensional shapes [30].

## 5 FUTURE WORK

When designing assistive technologies, it is important that the system can be adapted to people with varying abilities. As such, we would like to continue to investigate more UI modalities. One modality we have begun to investigate is using an augmented reality head-mounted display (HMD) such as the Microsoft HoloLens 2 in place of the projector. An HMD would additionally allow the system to display visual information that is not mapped to a surface, such as a floating window that the user can interact with through eye tracking or gestures. However, HMDs present an additional challenge in accurately aligning the robot's perception with the user's view. In order for the HMD to overlay virtual elements accurately on the real world we need to have a precise transform from the HMD to the environment.

The calibration of the projector can be improved through automatic projector-camera calibration techniques [8, 17, 21, 29]. These calibration techniques are faster, enable greater precision, and are capable of modelling distortion. Our current manual calibration method assumes that distortion is negligible and ignores distortion coefficients. Although distortion does not have a significant impact on our system, the size of the projected image could be increased with a wide-angle lens that might introduce significant distortion. This would introduce the need to correct for this distortion in our pipeline to maintain an accurate calibration when rendering the projected image.

## 6 CONCLUSIONS

We have presented the design of a user interface for an assistive robot manipulator that takes advantage of spatial augmented reality modalities. Object designation is achieved by projecting a joystick-operated cursor into the system's workspace. The workspace of the manipulator is communicated by highlighting surfaces that are within the manipulator's reach. The system can also highlight objects in order to communicate its intent to grasp an object.

Turning the world into the robot's interface through a projection system allows for deictic communication between the system and its user, by allowing both entities to point at objects for selection and confirmation. Such communication could be useful for other human-robot interaction domains where the interaction between people and robots includes objects in the world, either to augment or to replace gaze (e.g., [3, 4, 18, 24]) and pointing (e.g., [1, 2, 13, 16, 23]) used in prior research.

## ACKNOWLEDGMENTS

This work has been funded by the National Science Foundation (IIS-1763469). We would like to thank Michael Gonzales, Patrick Hoey, and Noah Torname for their contributions to the system.

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
