# OpenReview forum: "Spatial Augmented Reality User Interface for Assistive Robot manipulation"
_humanrobotinteraction.org/HRI/2023/Workshop/VAM-HRI — VAM-HRI 2023 Oral_

### Official Review · Program_Chairs · 2023-02-25
**Accept**

**Rating:** 7
**Confidence:** 5

**Review:**

Review 1:

This work investigates the design and development of an assistive robot manipulator mounted to a mobility scooter. This system intended to help users identify and confirm objects selected for manipulation. The authors do this by projecting colors directly into the user's environment to designate objects, intent communication, or the workspace. The authors then outline their current user study and plans to improve the system through an HMD and automatic projector-camera calibration.

Strengths:

Intended Users: The authors build a system for users who need mobility assistance in their daily life. It is always good to see these types of target users and fits the HRI theme “HRI for All.”

System Design: The authors clearly explain the design of their system and provide two feasible goals for future improvements.

Weaknesses:

Questionnaires: It is unclear why the Revised PSVT:R is used in this study. Is it because the users need to switch perspectives to the GUI which provides the objects with a different rotation? Also, the authors state the cognitive load may differ given the GUI vs. SAR UI. How will this be measured? A common measure for cognitive load is the NASA-TLX [1]

Notes:
Other papers that are relevant to this work that have not been cited are Morita et al. [2] and Zolotas et al. [3].

The authors will benefit from talking about their current and future work at VAM-HRI, therefore I would argue for it to be accepted.

[1] Hart, S. G., & Staveland, L. E. (1988). Development of NASA-TLX (Task Load Index): Results of empirical and theoretical research. In P. A. Hancock & N. Meshkati (Eds.), Human mental workload (pp. 139–183). North-Holland. https://doi.org/10.1016/S0166-4115(08)62386-9

[2] Kohei Morita et al., 2020. Extension of Projection Area using Head Orientation in Projected Virtual Hand Interface for Wheelchair Users. In 2020 59th Annual Conference of the Society of Instrument and Control Engineers of Japan (SICE). IEEE, 421-426. https://doi.org/10.23919/SICE48898.2020.9240271

[3] Mark Zolotas, Joshua Elsdon, and Yiannis Demiris. 2018. Head-mounted augmented reality for explainable robotic wheelchair assistance. In 2018 IEEE/RSJ International Conference on Intelligent Robots and Systems (IROS). IEEE, 18231829. https://doi.org/10.1109/iros.2018.8594002’

-----

Review 2:

In this paper, the authors present the design of a spatial augmented reality system that coordinates with a mobility scooter. The main components of the system involve the human designating an object to grasp, highlighting the robot's intended object, and displaying the robot's possible workspace. While no results are presented, the authors outline a user study in progress and their intended outcomes. I appreciate the careful design considerations discussed and the synthesis of related work.

I agree with Review 1's notes about carefully considering which questionnaires will be most useful for the user study. One other suggestion to improve the paper is to provide a figure with a system overview, to help readers visualize the system provided in section 3.

---

### Decision · Program_Chairs · 2023-03-02

Accept (Oral)